# A STEAM Experience in the Mathematics Classroom: The Role of a Science Cartoon

Daniela Marques [1], Teresa B. Neto [2,3], Cecília Guerra [2,3], Floriano Viseu [4,5], Ana Paula Aires [2,6,*], Marina Mota [2,3] and Ascensão Ravara [7,8]

1 Escola Secundária Romeu Correia, 2814-501 Almada, Portugal
2 CIDTFF—Research Centre on Didactics and Technology in the Education of Trainers, University of Aveiro, 3810-193 Aveiro, Portugal
3 Department of Education and Psychology, University of Aveiro, 3810-193 Aveiro, Portugal
4 Department of Integrated Studies on Literacy, Didactics and Supervision, University of Minho, 4710-057 Braga, Portugal
5 CIEd—Research Centre on Education, University of Minho, 4710-057 Braga, Portugal
6 Department of Mathematics, University of Trás-os-Montes e Alto Douro, 5000-801 Vila Real, Portugal
7 CESAM—Centre for Environmental and Marine Studies of the University of Aveiro, 3810-193 Aveiro, Portugal
8 Department of Biology, University of Aveiro, 3810-193 Aveiro, Portugal
* Correspondence: aaires@utad.pt

**Abstract:** A multidisciplinary team collaborated on the development of a learning experience involving 10th grade students using a Science, Technology, Engineering, Arts, and Mathematics (STEAM) approach. The experience was based on the development (conception, implementation, and evaluation) of a science cartoon that aimed to highlight different scientific and technological dimensions related to the diversity of marine worms (Phylo Annelida, class Polychaeta) present in the continental shelf off the coast of Aveiro, Portugal (NE Atlantic coast). The study was implemented in a Portuguese high school in the Aveiro region, involving 24 10th grade students, emphasizing a social context close to the students' lives. All pedagogical interventions occurred in face-to-face sessions during the 2020/21 school year and were oriented by the following research question: What is the role of science cartoons in establishing STEAM connections for solving real-world problems presented to 10th grade students? Following a qualitative and interpretative research methodology, with a design-based research focus, data were collected through a questionnaire, observations, and students' written records. The content analysis shows that most students learned new concepts related to STEAM areas. Evaluating the impact of the science cartoon reveals that it can be considered an innovative science communication resource due to its educational potential in stimulating a STEAM approach within the students' learning process.

**Keywords:** secondary school curriculum; mathematics education; science education; cartoon; STEAM

## 1. Introduction

According to Hursh et al. [1], education should be more than a diverse knowledge base from which students jump forward to areas of specialization. Those authors argue that the political approach to curriculum design ignores some very important aspects of cognitive development. One of these is the ability to integrate, or at least organize, knowledge and skills from different disciplines. Williams et al. [2] propose three concepts for integrating multiple fields of knowledge: interdisciplinarity, multidisciplinarity, and transdisciplinarity. Interdisciplinarity refers to the collaboration between two or more disciplines to address a particular problem or topic; multidisciplinarity involves the integration of knowledge from different disciplines without necessarily establishing connections between them Transdisciplinarity goes beyond the integration of disciplines and seeks a balanced relationship among the various disciplines, meaning that one discipline does not take priority over

another [2–4]. Successive reforms of curricula (e.g., in Portugal) have tended to value the connection of what is learnt in one subject (e.g., Mathematics) with what is learnt in other areas of knowledge (e.g., Biology). This complies with the concept of interdisciplinarity. In fact, specialists of different scientific disciplines (e.g., Design, Biology, Mathematics, Education) can demonstrate tolerance of and trust in each other's knowledge in solving problems (e.g., modelling problems). Interdisciplinarity, as opposed to monodisciplinarity, is thus gaining importance in the development of students' learning in diverse disciplines. In this vein, the "Students' Profile by the End of Compulsory Schooling" (SPECS) and the "Essential Learning Competencies" (ELC) proposed by the Portuguese Ministry of Education highlight the importance of interdisciplinarity to promote students' transversal competencies, such as the ability to conduct autonomous and collaborative work [5]. Open dialogue between different areas of knowledge is expected to be a great advantage for the labour market context.

In light of the importance attributed to interdisciplinarity and considering that knowledge can become more meaningful to students when it is applied in and outside mathematics education, mathematics teachers' professional knowledge could be developed in teacher training courses that are enhanced by the convergence of various disciplinary areas (e.g., Art, Technology). With this aim, some mathematics teacher training courses seek to promote the development of student teachers' pedagogical competencies (initial and in-service) under an interdisciplinary approach [3].

Currently, on a global scale [6,7] but also in the Portuguese context [8], we are witnessing the STEAM (Science, Technology, Engineering, Art, and Mathematics) movement in several educational contexts, such as mathematics education. The STEM (Science, Technology, Engineering, and Mathematics) approach has been studied by different authors internationally, such as Quigley et al. [6], Ejiwale et al. [7], and Guerra et al. [8]. However, Quigley et al. [9] go beyond the acronym STEM by adding Art, and they focus on the interplay of individual STEAM subjects, conceptualizing STEAM as a transdisciplinary instructional approach drawing on discipline integration, characteristics of the classroom environment, and problem-solving skills. According to Solin [4], Art projects could provide a favourable context to attract to mathematics diverse students who otherwise would not be interested. The same author advocates that an integration of STEAM areas in mathematics education can be a way of exposing the artistic aspect of mathematics content using simple visual coding or 3D modelling projects. Note that many artistic patterns (e.g., mandalas, mosaics) start with a basic shape or line, which can then be repeated, rotated, and combined with other shapes to form beautiful and complex designs. In fact, Quigley et al. [6] state that using a STEAM approach in learning activities can promote students' active learning about mathematics. However, regardless of curricular recommendations for the initiation of interdisciplinarity in education, some studies have stressed that the inclusion of an interdisciplinary approach in mathematics teaching practices is still limited [9]; thus, it is important to invest in more studies in this area.

The research presented in this paper was carried out by a multidisciplinary team during the 2020/2021 academic year within the framework of the project EmpowerScienceEDU ("Empowering science communication in educational research: a path for sustainable innovations in education") included in the "Scientific Initiation Programme" for Students in Education (PIC-Edu) led by the Research Centre on Didactics and Technology in the Education of Trainers (CIDTFF) at the University of Aveiro(PIC-Edu is a scientific initiation programme that promotes the integration of students in educational research. The research experience works on an annual basis, covering two academic semesters with a duration of 50 h per semester. The EmpowerScienceEDU project is grounded in the assumption that science communication can have a crucial role in promoting students' understanding of science and technology innovations.

The team was composed of one researcher in Education (coordinator of the EmpowerScienceEDU project), three teacher educators with expertise in mathematics education and supervision, one master's student in pre-service teacher (PST) training who was doing a

pedagogical internship in a Portuguese high school (Supervised Teaching Practice—STP), one designer and doctoral student in education, and one researcher in biology (curator of the Biological Research Collection (CoBI) and co-author of the scientific paper upon which the science cartoon in this study was developed).

The team conceived an innovative STEAM learning experience through the production of a science cartoon entitled "Diversity of polychaetes off the coast of Aveiro" [10]. The science cartoon was based on a scientific paper published in the field of biology entitled "Polychaeta (Annelida) from the continental shelf off Aveiro (NW Portugal): Species composition and community structure" [11].

The science cartoon aimed to highlight different scientific and technological dimensions related to marine biology issues. It translates (in illustrations and text dialogues) scientific knowledge related to the diversity of marine worms (polychaetes) existing on the continental shelf off Aveiro, Portugal (NE Atlantic coast).

## 2. Rationale

The STEAM approach has been studied by the scientific community of mathematics education [12] as well as science education [13] as a possible response to students' lack of interest in science, technology, engineering, and mathematics areas.

For Babette [14], there was a great need from both researchers and teachers in STEAM fields to stimulate students towards the STEAM approach and to realize that it makes sense to promote pedagogical creativity. Hence a new question arises in research, "How can we introduce creativity into classroom practices?"

In order to answer this question, various researchers (e.g., Conradty and Bogner [15]) have developed learning strategies and resources to enhance student creativity and motivation towards learning STEAM contents.

The STEAM concept has a more complete educational approach than STEM. The letter "A" added to the acronym STEM refers to "Art" and it arises from the perceived need to increase students' involvement with creativity and innovation in order to empower them with more transversal competencies for their future [15].

However, some doubts can be raised about what is considered Art within the STEAM acronym, i.e., whether we aim at an education of only "visual art" or of a "global perspective", which can include everything from visual to performing arts [16]. This idea is reinforced by other studies reporting that students with low academic achievement but an interest in the arts may become better students [17].

To assist teachers in the implementation of STEAM activities, Quigley et al. [6] have designed a model identifying concepts that students can use across disciplines and which enable them to learn something new by doing so (Figure 1).

They present the following explanation of the dimensions of the model:

-   discipline integration is when the teacher frames different areas or disciplines within a learning activity;
-   the classroom environment dimension involves the teacher's analysis of the classroom environment so that the moments become more conducive to learning, facilitating the resolution of the problem;
-   problem-solving skills are related to the feedback the teacher gives to students to support and develop their cognitive, interactional, and creative skills, depending on the activities.

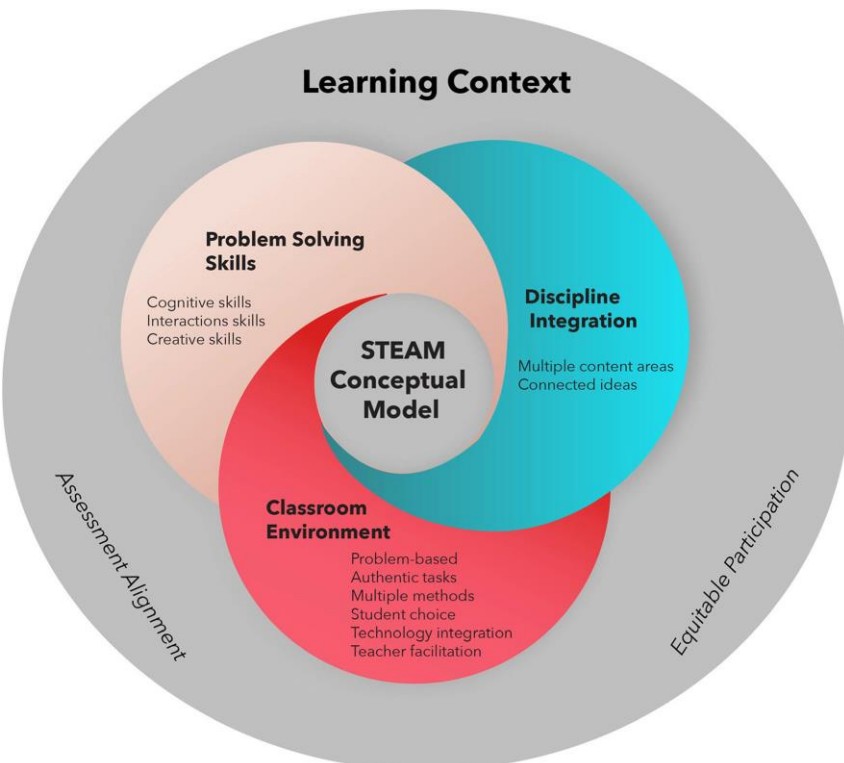

**Figure 1.** STEAM conceptual model [6] (p. 502).

For Costantino [18], having an interdisciplinary approach in STEAM education is an asset for both the student and the teacher, since it allows the student to perceive the problem from different perspectives and to understand that different perspectives can be applied in everyday contexts.

In designing STEAM learning activities, the inquiry-based learning process [15] can help to promote and motivate students' learning. This process begins with the exploration of a real-life problem which is simplified, structured, and idealized, turning it into a mathematical model. This process may integrate learning tasks, assessments, resources, environments, and teaching strategies, and it may reflect a commitment to student-centredness and learner empowerment while not neglecting individual knowledge. This process challenges individuals' everyday ideas about reality [14]. According to Perignat and Katz-Buonincontro [16], "Arts Education includes, but is not limited to, visual arts (such as drawing, painting, photography, sculpture, media arts, and design), performing arts (such as dance, music, and theatre), creative writing/poetry, expressive arts and crafts, digital and graphic arts, and design" [p. 34]. The integration of an interdisciplinarity concept in this study involves the articulation between "Arts", and Science, Technology, Engineering and Mathematics (STEAM), disciplines that are sometimes considered to be antagonistic from an epistemological point of view. Basically, the interesting challenge here is not whether "Arts" and STEM are different areas of knowledge, or if the students have learned about Arts/Mathematics/Biology, but rather, in what way their potentialities can help to think of new educational and didactic scenarios, particularly in mathematics teaching and the learning process.

In this sense, science comics can thus be implemented in STEAM activities because they are a didactic tool to communicate and teach science. The term "science comics" was introduced by Tatalovic [19] to define non-fiction-themed comics whose story plot is based on scientific facts. One of the main benefits of science comics could be the potential to communicate scientific topics through the visualization of complex concepts, often using metaphors and associating them with a narrative related to everyday objects and

experiences, helping the target audience to engage with the information at a personal level [20,21].

## 3. Method

We carried out a qualitative study with an interpretative focus [22,23] in order to evaluate the role and potential of a science cartoon as an instrument to establish STEAM connections for solving real-world problems presented to 10th grade Mathematics students. To this end, we implemented a design-based research (DBR) methodology [24–26] with the following specific aims:

- to address a problem through collaboration (including a multidisciplinary team of researchers, teachers, and students);
- to develop a practical solution to the identified problem (integrating design principles in the production of the science cartoon and its pedagogical exploration);
- to carry out evaluations to refine the proposed solution.

DBR methodology involves developing innovative pedagogical interventions in real contexts and includes a sequence of cycles of conception, implementation, and critical reflection [25,26].

Inspired by the DBR phases proposed by Reeves [25], our study was developed in 3 phases (Figure 2). The first phase (science cartoon design) involved the identification, selection, and analysis of the scientific knowledge to be considered in the elaboration of the pedagogical resource. The second phase (pedagogical intervention process) included carrying out the pedagogical intervention integrating the science cartoon in a school context and evaluating the science cartoon after students' comments. Finally, the third phase (science cartoon evaluation) was dedicated to evaluating the science cartoon's potential for promoting a STEAM approach. Based on the results emerging from second and third phases a final version of the science cartoon was produced. Each phase will be presented in detail in the following sections.

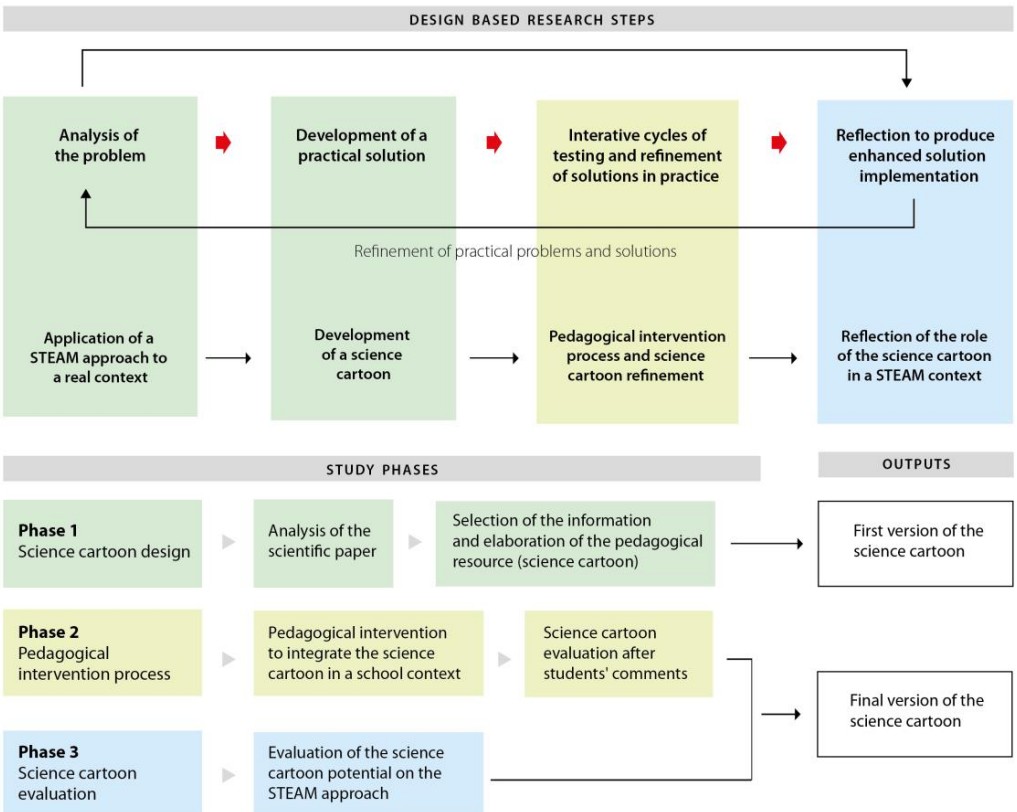

**Figure 2.** Phases of the design-based research methodology applied in the study (adapted from [25]).

*3.1. Phase 1—Science Cartoon Design*

The design of the science cartoon was based on the scientific paper [11],

- Ref. [11] that presents the results of a study on the diversity, abundance, and bathymetric distribution of marine invertebrates belonging to the class Polychaeta (Phylum Annelida) collected during an oceanographic expedition that took place in a limited area between the coast off Aveiro and the edge of the continental shelf. During the expedition, around 10,353 living organisms were found at different depths, allowing for the identification of 136 different species, which reflects the huge biodiversity hidden in the seabed.

After analysing the paper, a science cartoon was designed to communicate some of its scientific contents, which would later be addressed with more detail in the proposed mathematics exercises in the classroom. Thus, the science cartoon entitled "À descoberta dos poliquetas ao largo de Aveiro" (Discovering the polychaetes off the coast of Aveiro) [21] would also serve to introduce and frame the theme that was the basis for those exercises.

The ideation and production of the science cartoon took place between April 2021 and May 2021, following several meetings between the designer and the master's student to clarify different aspects regarding the appropriateness of the connection and the integration of different STEAM areas: Science, Technology, Engineering, Arts and Mathematics. The first version of the science cartoon (Figure 3) reflects the integration of all these areas.

The science cartoon translates (through images and text) the research process involved in the collection of marine invertebrates off the coast of Aveiro (NW Atlantic Ocean) using a sampling grab and highlights the diversity of the polychaete species, both locally on the continental shelf off Aveiro and globally on the planet. Later (in the final version of the cartoon), the sampling grab (the sampler used to collect the biological material) was replaced by the representation of a Remotely Operated Vehicle (ROV), a device also used to collect images and specimens from the sea floor. This allowed the use of geographic coordinates and depth data from the different sampling sites for programming a robot (in the classroom) using the Python language, defining a scientific exploration trajectory that simulates the ROV operation at sea.

The science cartoon thus mobilizes different knowledge, notably:

- Science (Biology): scientific data on the diversity and distribution of species of Polychaeta (Phylum Annelida) along a depth gradient from the coast to the shelf break (approximately 10 to 200 m depth), as well as their taxonomic relationships.
- Technology and Engineering: the model and programming of the robot that simulates the remote-operated vehicle used to collect/register the polychaete specimens;
- Art: the line-drawing representation of a biological research expedition and the morphological diversity of life forms;
- Mathematics: statistics studies involving the organization and treatment of data on species abundance;
- Sustainability: challenges related to the ocean preservation (e.g., calling attention to litter found on the ocean floor).

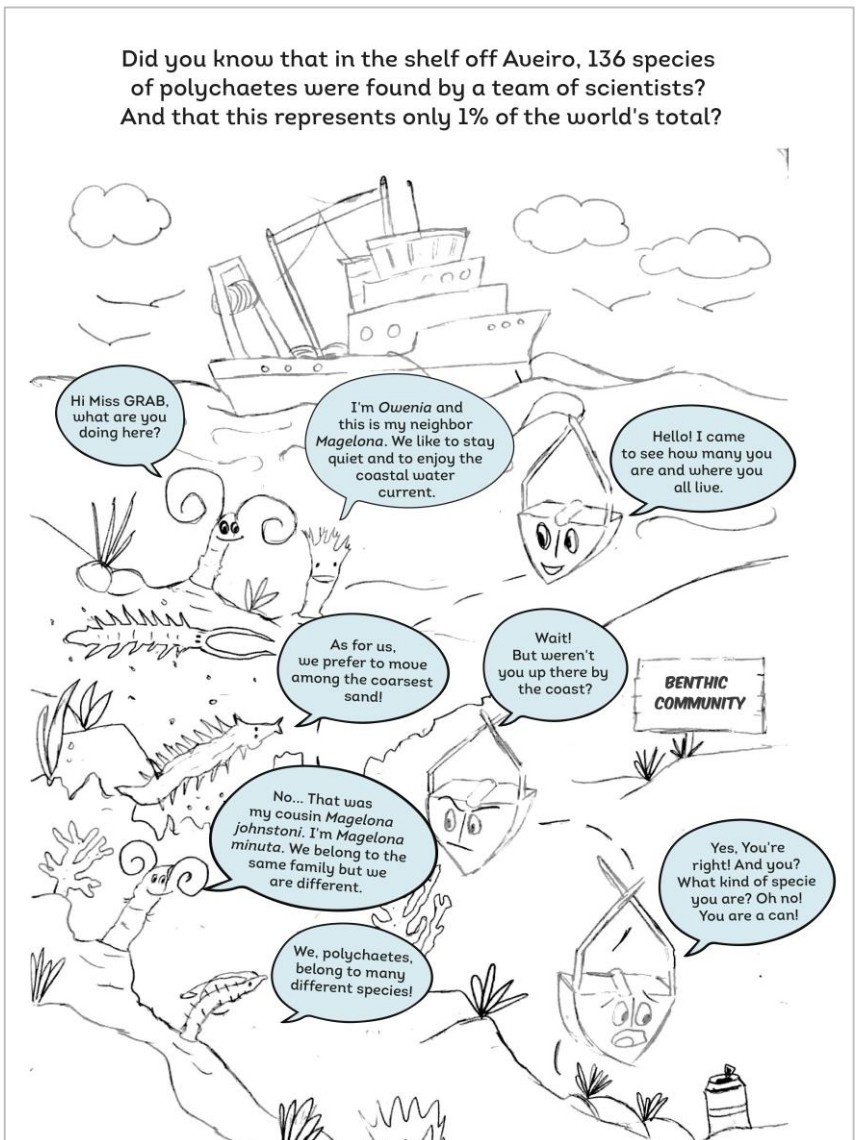

**Figure 3.** First version of the science cartoon. The polychaete species represented are (from left to right): *Magelona johnstoni* (Fiege, Licher & Mackie, 2000); *Owenia fusiformis* (Delle Chiaje, 1841); *Pisione remota* (Southern, 1914); *Eulalia mustela* (Pleijel, 1987); *Magelona minuta* (Eliason, 1962); and *Prionospio aluta* (Maciolek, 1985).

*3.2. Phase 2—Pedagogical Intervention Process*

Given the available resources (scientific article and metadata), the master's degree student decided to frame the exploration of the science cartoon within the scope of her practice component of pre-service teacher training (Supervised Teaching Practice—STP).

The science cartoon was explored within the subject of Mathematics with 24 10th grade students aged 15 and 16 years old (11 female and 13 male) from a Portuguese high school in the Aveiro region. The class integrates the Socio-economic Sciences track at high school and was considered homogeneous in terms of academic achievement. The curricular content selected was the "Organization and Treatment of Data" (Expected Learning in Mathematics of the 10th grade). All pedagogical interventions took place in face-to-face sessions during the 2020/21 academic year.

A mathematics learning activity was conceived using a STEAM approach, taking into account the three dimensions discussed above: problem solving, discipline integration, and classroom environment [8]. For that, three tasks were designed, following the principles of the STEAM approach (described in the following). The tasks were implemented in

3 sessions of 50 min each. The fact that the students were on the Socio-economic Sciences track of high school is considered to have made the tasks more challenging since most students did not have any particular affinity for the sciences.

### 3.2.1. Task 1: Exploration of the Science Cartoon

The first activity consisted of interpreting the science cartoon. For that, students had to answer the questions in a script (Appendix A—Task 1) aimed at helping them to:

- Interpret the title and content of the cartoon (questions (a) and (b));
- Identify areas of knowledge present in the cartoon (question (c));
- Suggest changes to be made to the cartoon in order to improve it (question (d)).

In question (a), the students were expected to understand from the cartoon title that the activity was related to marine exploration, even if they did not previously know the term "polychaetes". To answer questions (b) and (d), students needed to analyse and interpret the whole cartoon including title, speech bubbles, and the visual representation itself. Question (c) aimed to understand the depth of the students' analysis of the science cartoon by identifying the areas of knowledge highlighted in it. Table 1 summarizes the scientific terms that students were expected to detect and whose meaning they attempted to infer through their analysis of the cartoon.

**Table 1.** Pedagogical exploration of the science cartoon.

| Term | Definition or Function | Evidence in the Science Cartoon |
|---|---|---|
| Off Aveiro | An area at sea near Aveiro | The image of the ship at sea |
| Polychaetes | Marine organisms that live on the ocean floor at different depths | The inclined plane of the seabed and the representation of organisms along it |
| | Diversity of species (morphological forms and life styles) | The organisms' speech: "we Polychaetes belong to many different species"; "we like to stay quiet"/"we prefer to move among the coarsest sand". |
| Grab | An engine used to collect the polychaete specimens | By the grab's speech: "I come to see how many you are and where you live" |
| Benthic community | Community of organisms that live in association with the sediment on the ocean floor. | By the sign fixed to the bottom of the sea |

The answers given by the students to the script for interpreting the science cartoon were later used to improve the cartoon.

### 3.2.2. Task 2: Statistical Activity

In this task, a table with information about the number of individuals for each species per sampling station (taken from the scientific paper) was produced by the master's student (Appendix A—Task 2) and presented to the students. The students had to interpret and analyse the data statistically in order to sort the different polychaete species by depth. This task introduces ecological concepts such as bathymetric distribution, assemblages of species, and biological community structure.

### 3.2.3. Task 3: Problem-Solving Using Python Software

The last task aimed to initiate students into programming by simulating a possible route that the ROV used to collect the different species. A problem-solving task was set with two questions for students to programme in Python on a graphic calculator and calculator ecosystem (robot and Hub) (Appendix A—Task 3). This task helped the students to understand the process of maritime expedition in the discovery of marine invertebrate species.

There was only one instruction required: to travel in a straight line for 150.1 m. Given the spatial dimensions of the classroom, students worked with shorter measurements and

we adopted 15.01 dm as the required distance. The unit of measurement considered was always the decimetre as this is the Rover's unit of measurement.

### 3.3. Phase 3—Science Cartoon Evaluation

This phase aimed to understand the role of the science cartoon in establishing STEAM connections within a mathematics learning activity in the 10th grade classroom and to establish which STEAM knowledge areas were identified in the science cartoon during the learning activity. In this context, the evaluation of the science cartoon's potential to promote a STEAM experience required the collection of data regarding several aspects of this resource. The collection of data involved:

- The application of a questionnaire to the students;
- Classroom observations taken by the master's degree student;
- Document analysis of the students' written responses to the script.

Based on these data collection instruments, content analysis [27] allowed us to understand the relationship between the STEAM experience enriched by the science cartoon and the improvement of several student learning outcomes. In the next section, this paper will discuss the data gathered through the questionnaire, classroom observation and reflexive task 1.

## 4. Results

This section presents the results from the evaluation of the pedagogical role of the science cartoon and reveals which STEAM knowledge areas were identified by students in the science cartoon.

### 4.1. Pedagogical Role of the Science Cartoon

The role of the science cartoon in establishing STEAM connections in a mathematics learning activity applied to a 10th grade class in the Socio-economic Sciences track of high school can be highlighted by the results from task 1 (see Table 1). This task involved answering four questions about the cartoon (Appendix A—task 1).

In question (a) concerning the cartoon's title (see Figure 3), students were expected to understand that the cartoon was about a scientific exploration at sea and the discovery of marine organisms (see Table 1). By analysing the answers given, we found that students' difficulty was related to their lack of knowledge of the term "polychaetes". An example of this is the following transcript of an interaction between the teacher and two students during the exploration of question (a) for Task 1, as well as the corresponding answers (Figures 4 and 5).

| Original transcription | Translation |
|---|---|
| a) O que te sugere o título do cartoon? Porquê?<br><br>*Sugere que alguém [crossed out] irá à pesquisa de Poliquetas [crossed out] ao longo da costa de Aveiro pois o título foi claro do objetivo do cartoon.* | a) What does the title of the cartoon suggest to you? Why?<br><br>"It suggests that someone will go looking for polychaetes along the coast of Aveiro since the title was clear as to the cartoon's purpose." |

**Figure 4.** Answer to question (a) of the cartoon analysis guide—task 1 (Student C).

| Original transcription | Translation |
|---|---|
| a) O que te sugere o título do cartoon? Porquê?<br><br>Se eu soubesse o que era "poliquetas" teria entendido o título. | a) What does the title suggest to you? Why?<br><br>"If I knew what polychaetes are, I would have understood the title." |

**Figure 5.** Answer to question (a) of the cartoon analysis guide—task 1 (Student D).

> Student A: What do we really have to answer in the first question?
> Teacher: We want to know what you think is the meaning of the title of the cartoon.
> Student B: But I don't know what polychaetes are!
> Teacher: That's exactly what I want you to answer: what you understood from the title, what you didn't understand and why you didn't understand it.

Of the 24 students, more than half gave a response like that in Figure 4, with the remainder stating that they did not understand because they did not know the meaning of the word "Polychaete" (Figure 5). Only after analysing the whole science cartoon would students have identified the polychaetes as the "bugs" represented in the image.

It is important to note that the role of the teacher was vital to exploring the science knowledge presented in the science cartoon. Nevertheless, some students did not pay attention to the speech in the balloons.

This resource allowed not only the communication of scientific information in Biology (presented in [11]), but also supported interpretation of the information using the potentiality of "Art" (in the form of a science cartoon) to translate, analyse, and interpret the problem-solving statement in an innovative way. Therefore, the science cartoon allowed us to contextualize the problem-solving situation and promoted the interpretation of scientific data translated in a science cartoon [10]. In fact, as mentioned by [15], using an inquiry-based learning process within a STEAM approach could be a way to effectively explore a real-life problem, in which the problem is simplified, structured, and idealized, turning it into a mathematical model. In this study, results show that inquiry-based learning helps students to formulate and/or answer questions about the science cartoon and to apply common research techniques (data analysis). Therefore, it was intended that students would display an attitude of "reading beyond the data", as also mentioned by [15], constituting a higher level of reading and comprehension than what was actually demonstrated by the students.

The second question of the guide (question (b)) aimed to collect information about the students' interpretation of the scientific information contained in the science cartoon. From the analysis of their answers, almost all of them indicated that the objective was to collect polychaete worms, and some students recognized the diversity of polychaete species along the continental shelf off Aveiro.

In the written records, students emphasized pollution found on the seabed and the great diversity of species, relying mainly on elements that are explicit in the cartoon and with which they are familiar. For example, in the answer shown in Figure 6, Student E mentions the theme of pollution as well as the variety of polychaete species. He also mentions the sampling grab, but at the level of an animated character and not the function it performs.

| Original transcription | Translation |
|---|---|
| b) Após uma análise cuidada do cartoon, quais as ideias que retiras? Justifica.<br><br>As ideias que retiro são que as espécies de poliquetas são parecidas entre si uma vez que a Draga confundiu duas delas e que a poluição marinha é uma realidade cada vez mais presente. | b) After a careful analysis of the cartoon what ideas do you draw from it? Justify.<br><br>"The ideas I have retained are that the species of Polychaeta are similar as the grab confused two of the species. Another idea is that marine pollution is increasingly present." |

**Figure 6.** Answer to question (b) of the science cartoon script (Student E).

Throughout the various questions, students demonstrated a generally low capacity for reasoning and argumentation, since their answers were mostly based on direct answers about different dimensions of the resource. It is important to note that it was the first time that the students had been faced with the analysis of a science cartoon which aimed to connect several knowledge areas (e.g., Biology, Ecology, Mathematics).

Table 2 shows the absolute frequency of the topics mentioned by the students in their answers (b) (what ideas did students take away from the cartoon). However, the results show that some students were unable to confirm the topics identified through the justifications provided in their responses.

**Table 2.** Absolute frequency of student responses to question (b) (what ideas did students take away from the cartoon).

| Categories of Response | $n_i$ |
|---|---|
| It consisted of a maritime expedition to the Aveiro oceanic coast | 1 |
| The species are distributed on the continental shelf according to sediment granulometry | 2 |
| There is a great diversity of polychaete species | 21 |
| Pollution affects the species | 13 |
| Different species are found at different depths | 1 |
| The worms are collected using a sampling grab | 3 |
| Several species belong to the same family | 3 |
| Other | 2 |

Table 2 highlights in bold some of the keywords that associate the subject of the cartoon to other areas of knowledge (question (c)), allowing an understanding of the role of the science cartoon in establishing STEAM connections, namely in the following areas:

- Mathematics: the terms "distributed according to granulometry" and "different depths" can be explored in the curricular content related with "Organization and Data Treatment".
- Science: the terms "polychaete species" and "family" (a hierarchical taxonomy level) can be explored in the curricular contents of Biology;
- Technology: the terms "sampling grab" and "ROV" can be explored in disciplinary areas such as "Technology education".

Regarding question (d) (what changes could be made to improve the cartoon), some students suggested that the title of the science cartoon should be changed and argued that they were unfamiliar with the term polychaetes, so it was not so easy for them to understand the message of the science cartoon based on the title alone. Student F emphasized that some of the scientific information integrated into the polychaetes' speech could be simplified to make the meaning of expressions such as "polychaetes" more understandable (Figure 7). This argument was also presented by three other students.



| Original transcription | Translation |
|---|---|
| d) Que mudanças poderias introduzir no cartoon no sentido de melhorar a mensagem?<br><br>*Acho que o desenho transmite uma boa explicação do que são os poliquetas, contudo, considero que o texto poderia conter uma linguagem mais clara.* | d) What changes could you make to the cartoon to improve the message?<br><br>"I think the drawing gives a good explanation of what polychaetes are, however, I think the text could contain clearer language." |

**Figure 7.** Answer to question (d) of the science cartoon script (Student F).

One student stated that it was a different way of learning, recognizing that it was possible to learn new concepts (in this case about the polychaetes and the diversity of species) through a new approach other than via formal teaching.

After exploring the science cartoon and before proposing Tasks 2 and 3, a short introductory text was presented which provided a definition of "polychaete" ("The marine invertebrates you are going to see are marine annelids, called polychaetes. They are called so because they have a lot of chaetae (hair-like structures)"), which some of the students had not understood from the cartoon.

Regarding task 2 (statistical activity), the first question aimed to perform a statistical analysis of the number of individuals of each polychaete species according to depth. The table (Appendix A—Task 2) was not arranged in ascending order of depth or number of species. Therefore, students had to understand that at a certain depth, there were certain individuals of the corresponding species and that species did not have individuals at all the different depths recorded.

The answer in Figure 8 represents the most common strategy of analysis used by the students regarding the division of the number of individuals according to the three platforms (n = 18 students), i.e., choosing to organize the data in a table.

**Original transcription**

1.1. Recolhe os dados e analisa-os consoante o a profundidade e o número de indivíduos de cada espécie.

| | PI (8 a 22 metros) | PM (31 a 79 metros) | PE (94 a 178 metros) |
|---|---|---|---|
| Nº de indivíduos | 220 | 1732 | 183 |

**Translation**

1.1 Collect the data and analyse them according to the depth and number of individuals of each species.

| | IP - Inner platform (between 8 and 22 meters) | MP - Medium Platform (between 31 and 79 meters) | EP - External platform (between 94 and 178 meters) |
|---|---|---|---|
| Number of individuals | 220 | 1732 | 183 |

**Figure 8.** Example of an answer to question 1.1 of task 2.

However, some students chose to highlight the most or least abundant species. In Figure 9, the student indicated which species had the greatest number of individuals and the depth at which they were recorded. This answer is incomplete because the student did not fully make use of the data they had at their disposal either in the interpretation

text or in the table. Another student reported that the number of individuals of the species generally decreased as the depth increased (Figure 10).

| Original transcription | Translation |
|---|---|
| 1.1. Recolhe os dados e analisa-os consoante o a profundidade e o número de indivíduos de cada espécie. A espécie dominante é a Protodorvillea com 492 poliquetas aos 40.9 metros de profundidade, na plataforma média. | Question 1.1 Collect data and analyse according to depth and number of individuals of each species. Student A - The dominant species is Protodorvillea with 492 polychaetes at a depth of 40.9 metres on the middle shelf. |

**Figure 9.** Answer to question 1.1 of task 2 (Student A).

| Original transcription | Translation |
|---|---|
| 1.1. Recolhe os dados e analisa-os consoante o a profundidade e o número de indivíduos de cada espécie. A medida que a profundidade aumenta a quantidade de indivíduos diminuem em maior parte da espécies, porém algumas espécies só se verificou o seu aparecimento em altas profundidades. | Question 1.1 Collect data and analyse according to depth and number of individuals of each species. Student B - As the depth increases the number of individuals decreases in most species. However, some species only appear at greater depths. |

**Figure 10.** Answer to question 1.1 of task 2 (Student B).

The use of different and flexible problem-solving strategies to solve the problem was a means to lead students to understand the question correctly. Results show that students used different strategies to solve question 1.1 (see Figures 9 and 10). According to Elia et al. [28], integrating flexibility in non-routine problem-solving could be a way to promote students' problem-solving competencies.

The class discussion allowed us to reflect on the absolute frequencies in the three different zones—Inner platform: between 8 and 22 m deep, characterized by finer sediments; Middle Platform: between 31 and 79 m, characterized by coarse sediments; and External platform: between 94 and 178 m, also characterized by fine sediments but with some degree of heterogeneity—and to assess which zone had the highest number of individuals (the middle platform).

Another question of Task 2 involved determining the range of depth interval where the species *Mediomastus fragilis* could be found. Only half the class gave the full answer with written justification (giving the range of the extremes and the calculation of the amplitude, which is the difference of the extremes) resulting in an amplitude of 150.1 m. However, in the class discussion, when asked about the meaning of 150.1 m, the students showed that they knew that "the species *Mediomastus fragilis* could be found from the beginning of the depth 12.7 m to 162.8 m".

Task 3 challenged the students to programme with Python to simulate a possible route that the ROV used to collect different species, with only one condition: to travel in a straight line for 150.1 m. Being a simulation, it was suggested that students use 15.01 decimetre (the measurement worked out was always in decimetres as this is the Rover unit of measurement). The groups, consisting of five students each, explored the functions of the calculator graphic and as required, constructed a path, with Python, based on 5 lines of code. The Figure 11 illustrates an example of a solution presented after several attempts to run the programme.

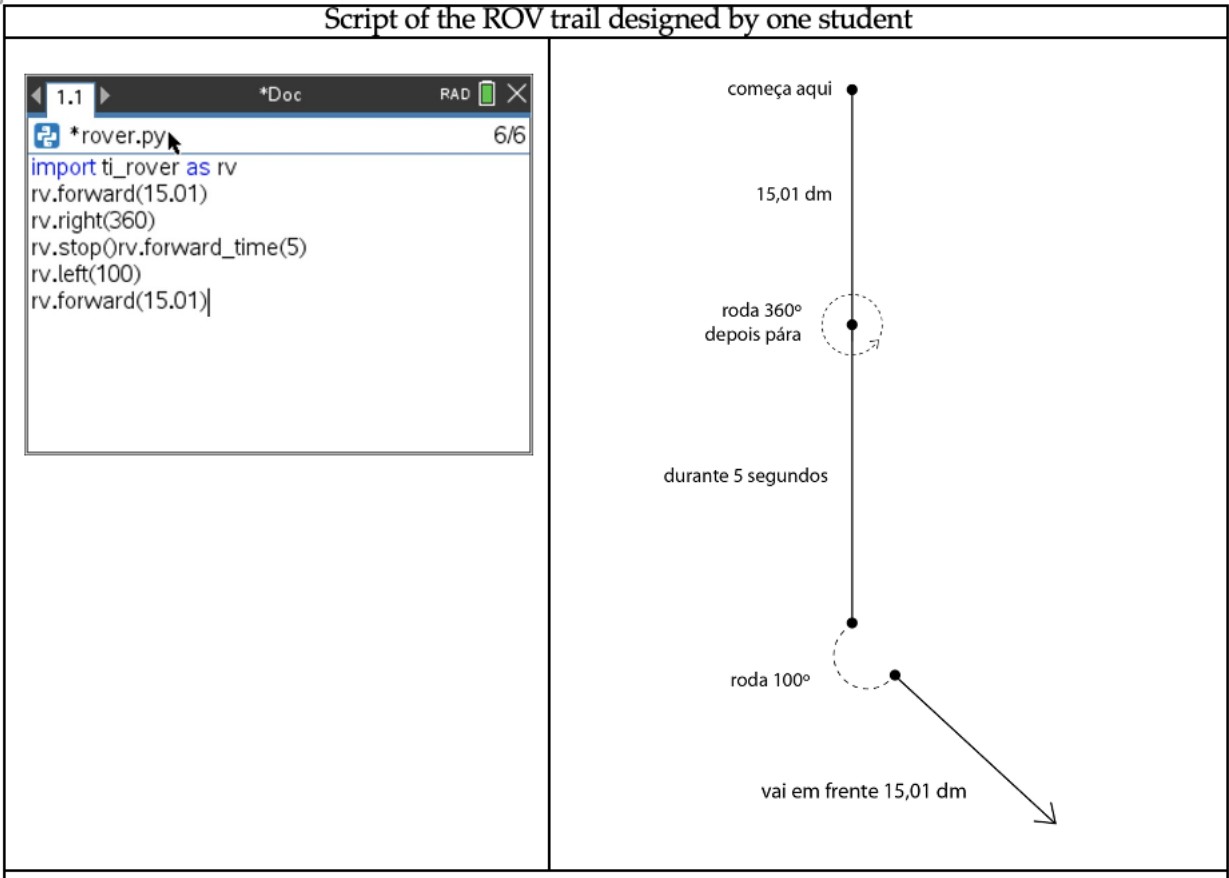

**Figure 11.** Answer by group 2 for Task 3.

The Python code supports the engineering design and simulation of the ROV's path to collect marine organisms. The students were encouraged to design their own path as researchers looking for different species of polychaetes.

*4.2. STEAM Knowledge Areas Identified in the Science Cartoon*

The application of the questionnaire after task 3 allowed us to collect students' perceptions on the pedagogical potential of the science cartoon. It is important to note that only 21 out of the 24 students provided responses to the questionnaire.

The feedback about the learning intervention collected from the students was very positive, with all 21 students who filled in the questionnaire (out of the 24 participants) reporting having learned new concepts, as shown in Table 3. The majority indicated that they had acquired knowledge referring to the existence of polychaete species, followed by knowledge of the diversity of these species off on the cost of Aveiro.

**Table 3.** Student responses to the question "I learned ... ".

| "I Learned ... " | $n_i$ |
| --- | --- |
| ... what polychaetes are | 13 |
| ... what STEAM means | 1 |
| ... to work with the Rover | 1 |
| ... to programme in Python | 1 |
| ... the diversity of polychaetes off the coast of Aveiro | 4 |
| ... how a sampling grab works | 1 |

This evaluation process allowed us to refine the Science Cartoon "Diversidade de poliquetas ao largo de Aveiro" [Diversity of polychaetes off the coast of Aveiro], with the final version presented below (Figure 12).

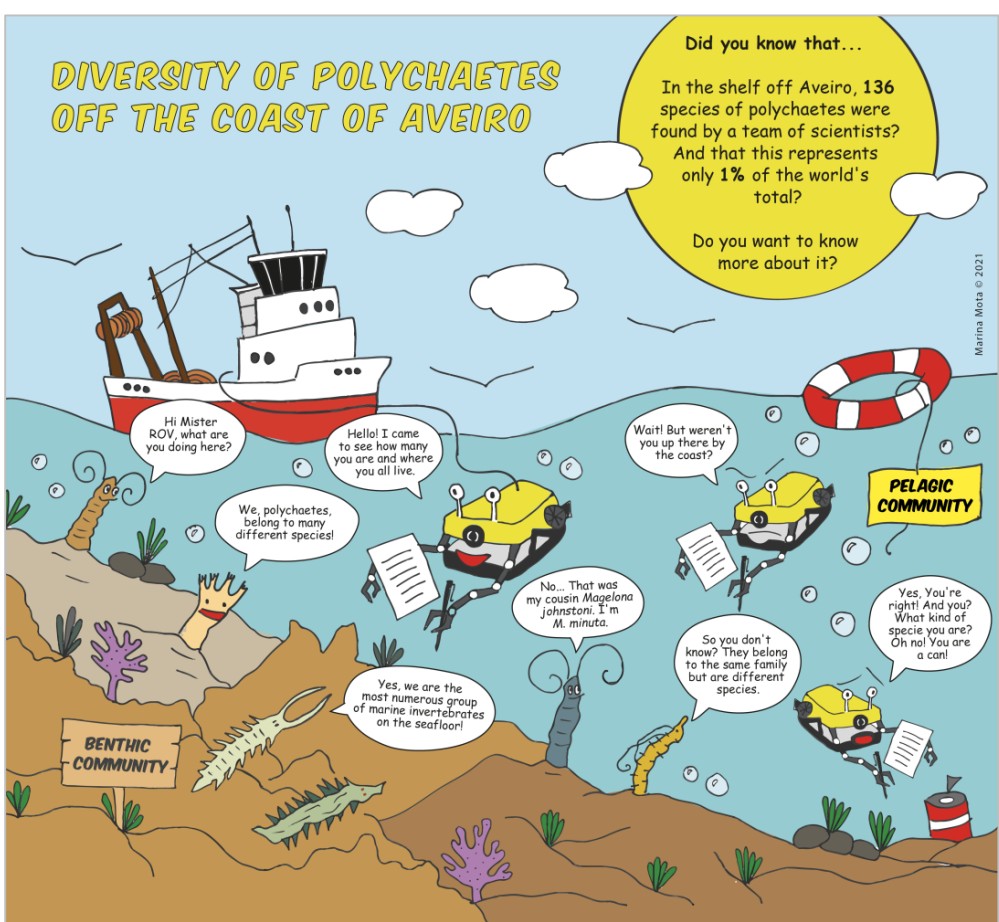

**Figure 12.** Final version of the science cartoon.

To celebrate the International Day for Biological Diversity (22 May 2021), the Biological Research Collection (CoBI), in partnership with the EmpowerScienceEDU Project, published this science communication resource on its social media channels, such as https://www.facebook.com/cobi.aveiro/ (accessed on 22 May 2021).

## 5. Conclusions

This study aimed to answer the following question: What is the role of science cartoons in establishing STEAM connections for solving real-world problems presented to 10th grade students?

The goal of the study was to evaluate the pedagogical potential of the science cartoon entitled "Diversity of polychaetes off the coast of Aveiro" [10] in the promotion of students' learning about curricular topics in 10th grade Mathematics.

In this study, the teacher used a science cartoon during a Mathematics class involving 24 10th grade students. The learning challenge was to explore, through a STEAM approach, a real-world problem related to a biology content, i.e., the identification of species of polychaetes in the proximity of the students' social context—the continental shelf off the coast of Aveiro, Portugal (NE Atlantic coast).

Data was collected and analysed to understand what the role of the science cartoon is in establishing STEAM connections in a Mathematics learning activity in the 10th grade classroom and to identify which STEAM knowledge areas participants recognized in the science cartoon during the learning activity.

Results showed that the science cartoon was a means to explore, interpret, and make sense of the information presented through images, identifying different STEAM areas. With the students' participation and comments, the final version of the science cartoon effectively translates the STEAM areas, specifically:

- Biology: the diversity of the species of Polychaeta (Filo Annelida) and their relationship with bottom depth;
- Arts: the drawn representation of the morphological diversity of polychaete forms;
- Engineering: the model of the robot that collects the polychaeta;
- Mathematics: statistics study involving the organization and treatment of data on species abundance;
- Sustainability: challenges related to ocean and seabed preservation (e.g., marine litter).

Most students claimed that through the science cartoon exploration they learned new concepts from other topics (e.g., Biology, Engineering).

The current recommendations for teaching and learning suggest the use of authentic problem-solving in interdisciplinary contexts, following an integrative STEAM approach [12,14–16,18]. Regarding the inquiry-based learning process [15], the students were given the opportunity to work in groups, to use different problem-solving strategies, and to justify their answers and propose new questions. This process gave students in a Mathematics class the opportunity to explore scientific information about Biology while using Arts in the form of a science cartoon [10].

The science cartoon, through images and text, played an important role in the completion of the learning activity, supporting the students' understanding of the process involved in the collection of marine invertebrates (Annelida, Polychaeta) using a sampling grab or a ROV, the diversity of species, and their main morphological characteristics and lifestyles. Nonetheless, this study showed that it is important to identify and support the adoption of working practices among teachers/researchers in order to promote effective integration of the various STEAM areas.

Future challenges to implementing a STEAM approach were identified from this study and include the need to promote a more active participation from students, namely in the definition of the problem situation, whose resolution involves the application of knowledge from various subject areas. It is important that students are engaged and empowered to develop by themselves the process of resolution while being able to share the process adopted. A possible path, taking the case presented here as an example, would

involve the collection of biological data, using Python programming in combination with statistical analysis of data collected for the study of polychaete communities, and with the dissemination of results through a cartoon. It seems to us equally important to promote forms of collaboration between teachers from different subject areas in the development of co-teaching situations, which translate the articulation of the STEAM areas in accordance with the curricular objectives and educational projects of the schools. In order to promote the "STEAM teacher" of the 21st century, it is crucial to start developing and reflecting this didactic approach in the initial stages of teacher education.

**Author Contributions:** Conceptualisation, D.M., T.B.N., C.G. and A.R.; methodology, D.M., T.B.N., C.G., A.R. and M.M.; formal analysis, D.M., T.B.N., F.V. and A.P.A.; investigation, D.M. and T.B.N.; writing—original draft preparation, D.M., T.B.N., C.G., F.V., A.P.A., A.R. and M.M.; writing—review and editing, D.M., T.B.N., C.G., F.V., A.P.A., A.R. and M.M.; supervision, T.B.N., F.V., A.P.A., C.G. and A.R.; funding acquisition, A.P.A.,T.B.N. and C.G. All authors have read and agreed to the published version of the manuscript.

**Funding:** This work is financially supported by National Funds through FCT—Fundação para a Ciência e a Tecnologia, I.P., under the project UIDB/00194/2020 (CIDTFF). C.G. and A.R. contract is funded by national funds (OE), through FCT, I.P., in the scope of the framework contract foreseen in the numbers 4, 5 and 6 of the article 23, of the Decree-Law 57/2016, of August 29, changed by Law 57/2017, of July 19; M.M. is funded by FCT through the individual research grant 2020.07278.BD; D.M. through the individual research grant BII/UI57/9608/2021.

**Institutional Review Board Statement:** Ethical review and approval were waived for this study due to data protection being assured by following the institutional protocol in place that regulates the practice component of PST training, including data collection procedures.

**Informed Consent Statement:** Informed consent was obtained from all participants by a form that was delivered to children's legal guardians.

**Data Availability Statement:** Data analyzed during this study are available from the first author on request.

**Conflicts of Interest:** The authors declare no conflict of interest.

## Appendix A. Description of Tasks 1, 2, and 3

Task 1: Cartoon analysis and exploration guide.

(a) What does the title of the cartoon suggest to you? Why?
(b) After a careful analysis of the cartoon, what ideas do you take away from it? Justify.
(c) How interesting is the content of the cartoon for your education? With what areas of knowledge (subjects) can you associate the content of the cartoon?
(d) What changes could you make to the cartoon to improve the message?

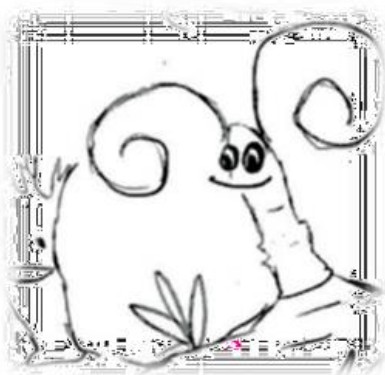

**Figure A1.** Representation of Magelona minuta on cartoon.

Task 2: Discovering the polychaetes of the continental shelf off Aveiro.

To explore and get to know a community of marine species, the marine invertebrates, existing on the Portuguese continental shelf, an oceanographic expedition was undertaken in an area limited between the coast and the edge of the continental margin shelf off Aveiro.

The marine invertebrates that will be observed are marine annelids, called polychaetes. They are so called because they have many different types of chaetae (hairs).

In the expedition carried out, about 10,353 invertebrate organisms belonging to the class Polychaeta (Phylum Annelida) were found (among others) at different depths, allowing the identification of 136 different species.

In order to help researchers, only 29 polychaete species were considered, representing a small sample of the total number of species. The attached table illustrates some of these species, as well as the number of individuals and the depth (in meters) in which they were found. We intend to perform statistical treatment of the data.

The aim of this study is to analyse the population size of the species in relation to the different depths recorded.

Use the table in Appendix B to answer the following questions:

The various species of marine invertebrates live at different depths. In addition to the biological material, sediment was also collected for granulometric analysis, which allowed the division of the platform into 3 zones with distinct characteristics:

Inner platform—between 8 and 22 m deep, characterised by finer sediments;

Middle Platform—between 31 and 79 m, characterised by coarse sediments;

External platform—between 94 and 178 m, also characterised by fine sediments, but with some degree of heterogeneity.

1.1 Collect the data and analyse them according to the depth and number of individuals of each species.

1.2 What is the percentage of individuals collected at a depth of 94 m or more?

1.3 *Mediomastus fragilis* is one of the species of which more individuals were found. At what depth range has it been recorded?

Task 3—Design the ROV path.

This task assumes that the polychaete species found are distributed along a straight line, starting at minimum depth (12.7 m) and ending at maximum depth (162.8 m).

1. Using the graphing calculator, we will build a programme in Python that simulates the polychaetes' sampling path using the Rover.

2. Now that you know the basic functions to make the robot move, imagine that you are the one to command the robot. Develop, at your discretion, a programme that gives other instructions to the Rover (for example: move in another direction, turn on a light).

**Appendix B. Adaptation of a Short Extract from the Original Data Matrix from Ravara and Moreira (2013) [11]**

| Species list/Depth | 157.7 | 76.8 | 162.8 | 31.8 | 59.1 | 131.5 | 134.9 | 71.7 | 34.9 | 57.3 | 46.1 | 31 | 94.1 | 39.3 | 48.3 |
|---|---|---|---|---|---|---|---|---|---|---|---|---|---|---|---|
| *Ampharete finmarchica* | 2 | | 2 | | | 1 | | | | | | | | | |
| *Amphicteis gunneri* | | | 1 | | | | | | | | | | | | |
| *Anobothrus gracilis* | 3 | | 1 | | | | | | | | | | | | |
| *Isolda pulchella* | 6 | | | | | | | | | | | | | | |
| *Sosane sulcata* | 4 | | | | | | 1 | | | | | | | | |
| *Chloeia venusta* | 1 | 9 | | 1 | | | 1 | | | | | | | | |
| *Mediomastus fragilis* | 1 | 40 | 2 | 7 | 41 | | | 74 | 7 | 1 | 24 | | | 2 | 4 |
| *Notomastus latericeus* | | 3 | | 15 | 13 | | 3 | 12 | 7 | | 15 | | | | |
| *Notomastus profundus* | 2 | | | | | | | | | | 2 | | | 7 | 5 |
| *Caulleriela alata* | | 1 | | | | | | | | | 3 | | | 1 | |
| *Caulleriella bioculata* | 7 | 1 | | | 1 | | | 2 | 1 | 1 | 4 | 2 | | 6 | 5 |
| *Chaetozone setosa* | 3 | | 1 | | | 6 | | | | | | | 5 | | 2 |
| *Cirriformia tentaculata* | | | 1 | | | | | | | | 1 | | | | 1 |
| *Protodorvillea kefersteini* | | 10 | 7 | 80 | 38 | 3 | | 21 | 93 | | 18 | | | | |
| *Schistomeringos neglecta* | | 1 | 2 | | 3 | | | | | | | | | | |
| *Eunice vittata* | | | | | | | | | | | | | | | 1 |
| *Marphysa bellii* | | | 1 | | | | | | | | 1 | | | | |
| *Diplocirrus hirsutus* | | | | | | | | | | | | | | | |
| *Pherusa monilifera* | | | | | | | | | | | | | | | |
| *Glycera lapidum* | | 13 | | 51 | 25 | | 1 | 19 | 10 | | 27 | | | | |
| *Glycera tridactyla* | 2 | 1 | 1 | | | | 1 | | | 1 | | 4 | 5 | 4 | |
| *Goniada emerita* | | 7 | | | | | | | 1 | | | | | | |
| *Goniadella galaica* | | | | 3 | | | | | 7 | | | | | | |
| *Goniadella gracilis* | | 5 | | | 16 | | | 19 | | 10 | 10 | 4 | | 13 | 13 |
| *Scoletoma cf. magnidentata* | | | | | | | | | | | | | | | |
| *Magelona alleni* | | | | | | | | | | | | | | 1 | |
| *Magelona filiformis* | | 1 | | | | | | | | | | | 5 | | |
| *Magelona johnstoni* | | | 1 | | | 6 | | | | | | | 2 | | |
| *Magelona minuta* | 1 | | 1 | | | | 6 | | | | | | 15 | | |
| total | 32 | 85 | 26 | 159 | 137 | 16 | 13 | 147 | 126 | 18 | 100 | 10 | 32 | 34 | 31 |

**Figure A2.** Number of individuals detected by depth.

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
