# Peer review of "A STEAM Experience in the Mathematics Classroom: The Role of a Science Cartoon"

_education, doi:10.3390/educsci13040392_

Round 1

Reviewer 1 Report (Previous Reviewer 3)

This study produces many positive results for students. In the conclusion section, please address the future challenges of implementing a similar STEAM lesson in the classroom. 

Author Response

 Comments and Suggestions for Authors

This study produces many positive results for students. In the conclusion section, please address the future challenges of implementing a similar STEAM lesson in the classroom. 

Following this recommendation, we insert at the end of the conclusions some implications of the study.

Reviewer 2 Report (Previous Reviewer 2)

Are lines 117-121 included in the text? if so they are a repeat of the previous sentence.

other than this the paper is very much more coherent and readable than the first version. The authors should be commended for the improvements they have made.

Author Response

 Comments and Suggestions for Authors

Are lines 117-121 included in the text? if so they are a repeat of the previous sentence.

We've already eliminated the 1st sentence: Nonetheless, some doubts can be raised regarding what is considered Art within the STEAM acronym, whether we aim for an education of only “visual art” or in a “global perspective” which can include everything from visual arts to performing arts [16].

We are grateful for the commendation, that are due, in large part, to the suggestions and contributions made by the reviewers.

This manuscript is a resubmission of an earlier submission. The following is a list of the peer review reports and author responses from that submission.

Round 1

Reviewer 1 Report

General Comments

This article presents a STEAM approach experience in the mathematics classroom with 10th grade Portugese students: the role of a science cartoon. The introduction and rationale are poorly written. Many important terms are not explained clearly such as transdisciplinarity, interdisciplinarity, STEAM approach and so on. A major rewrite is required. The rationale to conduct the study is not convincing. The analysis of data is superficial.

Specific Comments

1.     The author(s) should provide more elaborations regarding the notions of transdisciplinarity and interdisciplinarity. Providing examples would be helpful.

2.     Line 30-31, what do you mean by “interconnection of areas or disciplines without overlapping them”? Please give specific examples to illustrate your point.

3.     Should mathematics teacher training courses be conducted under the transdisciplinary approach?

4.      Line 39, what do you mean by “convergence of various disciplinary areas”?

5.     Please elaborate briefly how computational thinking can be helpful in any other context of studens’ lives (personal, professional and social).

6.     What is STEAM approach? Is it transdisciplinary or interdisciplinary or multidisciplinary?

7.     What do you mean by “both transdisciplinary and interdisciplinary concepts present in curriculum”? 

8.     What is the foundation of STEAM approach? Science and Mathematics?

9.     Why did the STEAM intervention implement in a mathematics classroom but not in other classroom such as a science classroom? If it is in a mathematics classroom then it is important to highlight what new mathematical concepts students learned through this intervention. Students used their mathematical skills to solve the problem but not learning new mathematical concepts. 

10.  Based on Table 3, no student indicates that s/he learns mathematics. Is this intervention appropriate to be implemented in a mathematics classroom? Have you considered implementing this intervention in a science classroom?

Author Response

We thank you for the support in the careful and thorough review for the improvement of our manuscript.

We have made the revision and we hope to have responded to the request.

  1. The author(s) should provide more elaborations regarding the notions of transdisciplinarity and interdisciplinarity. Providing examples would be helpful.

Considering the suggestion given, we have improved the text in the introduction and in the rationale focusing on interdisciplinarity.

  1. Line 30-31, what do you mean by “interconnection of areas or disciplines without overlapping them”? Please give specific examples to illustrate your point.

In order to clarify the sentence, we explain the meaning: This means that interdisciplinary learning requires interaction of knowledge from different disciplines to solve an overarching topic, theme, or problem by the students.

  1. Should mathematics teacher training courses be conducted under the transdisciplinary approach?

This is desirable and requires collaborative work between teachers from different areas.

This study was conducted by a multidisciplinary team.

  1. Line 39, what do you mean by “convergence of various disciplinary areas”?

About “Convergence of various disciplinary areas” we pretend means the meeting (intersection) of various areas.

  1. Please elaborate briefly how computational thinking can be helpful in any other context of studens’ lives (personal, professional and social).

The term computational thinking was deleted in the introduction because the focus of the paper is not this.

  1. What is STEAM approach? Is it transdisciplinary or interdisciplinary or multidisciplinary?

In this study we assume STEAM to be an interdisciplinary approach, in the sense of being a connection between several areas, without the prevalence of any of the areas.

  1. What do you mean by “both transdisciplinary and interdisciplinary concepts present in curriculum”? 

We reformulated this part of manuscript.

  1. What is the foundation of STEAM approach? Science and Mathematics?

The foundation is not just Science and Mathematics.

Taking into account the school we want for the 21st century, our concern was to develop in students the transversal skills in STEAM, assuming inclusive education as a fundamental principle, in order to meet the diversity of the needs of all students, in line with the principles of Universal Learning Design.

  1. Why did the STEAM intervention implement in a mathematics classroom but not in other classroom such as a science classroom? If it is in a mathematics classroom then it is important to highlight what new mathematical concepts students learned through this intervention. Students used their mathematical skills to solve the problem but not learning new mathematical concepts.

The first author, a trainee mathematics teacher, developed the study in one of her trainee classes. The Portuguese Mathematics curriculum for 10th grade involves the topic of Statistics. In this pedagogic intervention were proposed tasks applying concepts already studied in previous years. The new thing was the programming of a robot using Python language (technology).

  1. Based on Table 3, no student indicates that s/he learns mathematics. Is this intervention appropriate to be implemented in a mathematics classroom? Have you considered implementing this intervention in a science classroom?

The aim of this pedagogic intervention was to apply concepts of Statistics already studied in previous years, in the proposed tasks. For this question, we think that students considered that they did not learn new mathematical concepts.

Reviewer 2 Report

This is an interesting proposition but it is either poorly conceived or, possibly, just poorly described. Weak written English makes it difficult to judge. I attach a file in which I started to note specific instances of poor grammar but these were so frequent (almost every line) that I stopped after the abstract!

Early on the acronym STEM is used interchangeable with STEAM (e.g. citing STEM papers to support STEAM concepts) and this does not seem to be valid, especially as this particular project relies on some Art.

The exploration of a cartoon to convey scientific information and then use this as a basis for some programming is interesting. The research to see what, if anything, the pupils learned is not very robust, and the conclusions are therefore always going to be limited.  Sentences such as "In what concerns the science cartoon, it had a preponderant role in the accomplishment of the mathematical learning activity and for the understanding of the process involved concerning the collection of marine invertebrates (Polychaeta-Annelida) using a dredge or ROV, representing through images and text, the diversity of species and their main characteristics, including morphological ones" are almost meaningless: if this is trying to state that the cartoon enhanced the learning of the pupils then this would be overclaiming what the evidence supports. A better design would have been to present a similar group of pupils with some text conveying the same information and seeing which group learned the most. Given it is now too late to redesign the research, It would be better presented as a case study of student responses to the cartoon approach.

Author Response

We thank you for the support in the careful and thorough review for the improvement of our manuscript.

We have made the revision, responding to each of the comments. We hope to have responded to the request.

This is an interesting proposition but it is either poorly conceived or, possibly, just poorly described. Weak written English makes it difficult to judge. I attach a file in which I started to note specific instances of poor grammar but these were so frequent (almost every line) that I stopped after the abstract!

A linguistic revision of the entire text was carried out by a native speaker. Regarding the description the text has been significantly improved.

Early on the acronym STEM is used interchangeable with STEAM (e.g. citing STEM papers to support STEAM concepts) and this does not seem to be valid, especially as this particular project relies on some Art.

The focus of our work is STEAM and the text has already been clarified.

The exploration of a cartoon to convey scientific information and then use this as a basis for some programming is interesting. The research to see what, if anything, the pupils learned is not very robust, and the conclusions are therefore always going to be limited.  Sentences such as "In what concerns the science cartoon, it had a preponderant role in the accomplishment of the mathematical learning activity and for the understanding of the process involved concerning the collection of marine invertebrates (Polychaeta-Annelida) using a dredge or ROV, representing through images and text, the diversity of species and their main characteristics, including morphological ones" are almost meaningless: if this is trying to state that the cartoon enhanced the learning of the pupils then this would be overclaiming what the evidence supports. A better design would have been to present a similar group of pupils with some text conveying the same information and seeing which group learned the most. Given it is now too late to redesign the research, It would be better presented as a case study of student responses to the cartoon approach.

The text was revised in order to make the link between the definition of the study problem, the research objectives, the theoretical framework, the methodology, the results and the conclusions more coherent.

The scheme that we have now integrated into the manuscript (Figure 2) clarifies the development of the various phases that the study involved.

In promoting interdisciplinarity we did not focus on the prevalence of some of the areas of knowledge (STEAM), but rather on a dialogue between them.

Reviewer 3 Report

Many English grammatical errors are found (as shown in the Abstract for example). The manuscript requires thorough professional editing and proofreading and is not ready for review.  

Overwhelming grammatical errors undermine the quality and clarity of this manuscript.

Author Response

We thank you for the support in the careful and thorough review for the improvement of our manuscript

We have made the revision, considering your suggestion. We hope to have responded to the request.

Many English grammatical errors are found (as shown in the Abstract for example). The manuscript requires thorough professional editing and proofreading and is not ready for review.  

Overwhelming grammatical errors undermine the quality and clarity of this manuscript.

The manuscript has had a thorough professional editing and proofreading. A linguistic revision of the entire text was carried out by a native speaker.

Reviewer 4 Report

The study is interesting and significantly contribute the literature. However some part need improvement as mentioned below:

1. Reorganised your introduction. Explain why science cartoon, what is the current practise, problem in mathematics that leads to this study. Some parts need to replace as in methodology. Your argument flow need to be revised to show the sequence that leads to the study. 

2. Under the methodology part please elaborate how you conduct the study, why the sample was chosen, how long the process. Is inquiry learning as the contribution to the result? Why inquiry learning used in this study was unclear.

3. How does the STEAM conceptual impacted or used in the findings of the study?

4. The research question is not being answered well. Please revise it.

Author Response

We thank you for the support in the careful and thorough review for the improvement of our manuscript.

We have made the revision, responding to each of the comments. We hope to have responded to the request.

The study is interesting and significantly contribute the literature. However some part need improvement as mentioned below:

  1. Reorganised your introduction. Explain why science cartoon, what is the current practise, problem in mathematics that leads to this study. Some parts need to replace as in methodology. Your argument flow need to be revised to show the sequence that leads to the study. 

The introduction has been arranged, responding to the suggestions.

  1. Under the methodology part please elaborate how you conduct the study, why the sample was chosen, how long the process. Is inquiry learning as the contribution to the result? Why inquiry learning used in this study was unclear.

The methodology was clarified regarding methodological options, participants, study design, data collection methods and data analysis.

The scheme that we have now integrated into the manuscript (Figure 2) clarifies the development of the various phases that the study involved.

  1. How does the STEAM conceptual impacted or used in the findings of the study?

The results reflect the development of STEAM skills in students.

  1. The research question is not being answered well. Please revise it.

The research question has been clarified and we consider that it has been answered, taking into consideration the suggestions of the reviewers for which we are very grateful.

Round 2

Reviewer 1 Report

Comments

1.     It is still unclear to me the notions of transdisciplinarity, interdisciplinarity and multidisciplinarity. The authors should point to the exact location in the paper where they have addressed this query.

2.     The appropriateness of implementing the present intervention in a mathematics classroom is questionable. Bear in mind that the new thing that students learned in the intervention was the programming of a robot but not mathematics.

3.     The intervention also involved the application of other knowledge such as biology but why a mathematics lesson is chosen for the intervention but not a biology lesson?

Author Response

We thank you for the support in the careful and thorough review for the improvement of our manuscript.

We have made the revision and we hope to have responded to the request.

  1. It is still unclear to me the notions of transdisciplinarity, interdisciplinarity and multidisciplinarity. The authors should point to the exact location in the paper where they have addressed this query.

We clarify the notions mentioned with the following text:

Williams et al. [2] refer that interdisciplinarity, multidisciplinarity, and transdisciplinarity are three different ways of integrating multiple fields of knowledge: interdisciplinarity refers to the collaboration between two or more disciplines to address a particular problem or topic; multidisciplinarity, on the other hand, involves the integration of knowledge from different disciplines without necessarily establishing connections between them; and transdisciplinarity goes beyond the integration of disciplines and seek a balanced relationship among the various disciplines-meaning that one discipline does not take priority over the other.

We think it’s clearer.

  1. The appropriateness of implementing the present intervention in a mathematics classroom is questionable. Bear in mind that the new thing that students learned in the intervention was the programming of a robot but not mathematics.

In fact, the statistical mathematical content was not new. The novelty lies in the application of previously acquired statistical concepts in a real context problem, allowing students to treat and interpret data and make sense of it. The purpose of the intervention was to develop STEAM skills in a real context rather than to study a specific content of the curriculum.

  1. The intervention also involved the application of other knowledge such as biology but why a mathematics lesson is chosen for the intervention but not a biology lesson?

The first author, a trainee mathematics teacher, developed the study in one of her trainee classes.

The methodological guidelines of the Mathematics curriculum programs, for all levels of education, contemplate interdisciplinary work. In order for these guidelines to be effective, we consider it relevant to dynamize experiences according to a STEAM approach in initial teacher education.

Furthermore, the scientific article upon which the intervention was based includes an ecological data set (number of individuals of each species per location) ideal for statistical analysis. The cartoon was used here to contextualize the mathematical exercise presented to the students in the investigation carried out by the authors of the article, while introducing concepts from other areas, according to a STEAM approach.

Reviewer 3 Report

My concerns haven’t been addressed. 

“data were collected through questionnaire, observation and students’ written.”

This is an incomplete sentence.

“Content analysis emphasize that”

Content analysis emphasizes that

Author Response

We thank you for the support in the careful and thorough review for the improvement of our manuscript

We have made the revision, considering your suggestion. We hope to have responded to the request.

My concerns haven’t been addressed. 

“data were collected through questionnaire, observation and students’ written.”

This is an incomplete sentence.

The sentence was completed: “data were collected through questionnaire, observation and students’ written records.

“Content analysis emphasize that”

Content analysis emphasizes that

Thank you for the suggestion, but this sentence does not appear in this version.

The manuscript has had a thorough professional editing and proofreading. A linguistic revision of the entire text was carried out by a native speaker.
